# The phosphatidylinositol-5′ phosphatase synaptojanin1 limits integrin-mediated invasion of *Staphylococcus aureus*

Yong Shi,[1,2] Petra Muenzner,[1] Stefanie Schanz-Jurinka,[1] Christof R. Hauck[1,3]

**ABSTRACT** The gram-positive bacterium *Staphylococcus aureus* can invade non-professional phagocytic cells by associating with the plasma protein fibronectin to exploit host cell integrins. Integrin-mediated internalization of these pathogens is facilitated by the local production of phosphatidylinositol-4,5-bisphosphate (PI-4,5-$P_2$) via an integrin-associated isoform of phosphatidylinositol-5′ kinase. In this study, we addressed the role of PI-4,5-$P_2$-directed phosphatases on internalization of *S. aureus*. ShRNA-mediated knockdown of individual phosphoinositide 5-phosphatases revealed that synaptojanin1 (SYNJ1) is counteracting invasion of *S. aureus* into mammalian cells. Indeed, shRNA-mediated depletion as well as genetic deletion of synaptojanin1 via CRISPR/Cas9 resulted in a gain-of-function phenotype with regard to integrin-mediated uptake. Surprisingly, the surface level of integrins was slightly downregulated in Synj1-KO cells. Nevertheless, these cells showed enhanced local accumulation of PI-4,5-$P_2$ and exhibited increased internalization of *S. aureus*. While the phosphorylation level of the integrin-associated protein tyrosine kinase FAK was unaltered, the integrin-binding and -activating protein talin was enriched in the vicinity of *S. aureus* in synaptojanin1 knockout cells. Scanning electron microscopy revealed enlarged membrane invaginations in the absence of synaptojanin1 explaining the increased capability of these cells to internalize integrin-bound microorganisms. Importantly, the enhanced uptake by Synj1-KO cells and the exaggerated morphological features were rescued by the re-expression of the wild-type enzyme but not phosphatase inactive mutants. Accordingly, synaptojanin1 activity limits integrin-mediated invasion of *S. aureus*, corroborating the important role of PI-4,5-$P_2$ during this process.

**IMPORTANCE** *Staphylococcus aureus*, an important bacterial pathogen, can invade non-professional phagocytes by capturing host fibronectin and engaging integrin α5β1. Understanding how *S. aureus* exploits this cell adhesion receptor for efficient cell entry can also shed light on the physiological regulation of integrins by endocytosis. Previous studies have found that a specific membrane lipid, phosphatidylinositol-4,5-bisphosphate (PIP2), supports the internalization process. Here, we extend these findings and report that the local levels of PIP2 are controlled by the activity of the PIP2-directed lipid phosphatase Synaptojanin1. By dephosphorylating PIP2 at bacteria-host cell attachment sites, Synaptojanin1 counteracts the integrin-mediated uptake of the microorganisms. Therefore, our study not only generates new insight into subversion of cellular receptors by pathogenic bacteria but also highlights the role of host cell proteins acting as restriction factors for bacterial invasion at the plasma membrane.

**KEYWORDS** *Staphylococcus aureus*, internalization, integrin, phosphatidylinositol-4, 5-bisphosphate, phosphatase, synaptojanin1

Address correspondence to Christof R. Hauck, christof.hauck@uni-konstanz.de.

The authors declare no conflict of interest.

See the funding table on p. 17.

The gram-positive pathogen *Staphylococcus aureus* represents a major health concern due to increased antibiotic resistance (1–3). While *S. aureus* is generally regarded as

an extracellular pathogen, several disease manifestations and persistence of infection seem to be associated with invasive, intracellular bacteria (4–6). Indeed, *S. aureus* can invade diverse non-professional phagocytes, where it can escape into the host cell cytosol (7–9). A major route of entry for *S. aureus* relies on the exploitation of integrins, conserved adhesion receptors of mammalian cells (10, 11). Integrin α5β1 is abundantly expressed by various cell types and appears to be the major target of *S. aureus* and other integrin-engaging pathogenic bacteria (11–14). While some microbial pathogens, such as *Y. enterocolitis*, directly engage integrins via integrin-binding adhesins, *S. aureus* employs an indirect route to contact these receptors. In this regard, the staphylococcal fibronectin-binding proteins A and B (FnBPA and FnBPB) capture and immobilize the plasma proteins fibrinogen and fibronectin on the surface of *S. aureus* (15–17). Presentation of their protein ligands to integrins then triggers the efficient internalization of the ligand-decorated microorganisms by mammalian cells (16, 18, 19). Endocytosis of ligand-bound integrins requires the coordinated action of multiple cytosolic host factors, which orchestrate the actin cytoskeleton dynamics and membrane remodeling during this process (20). Besides protein tyrosine kinases (PTKs) of the Src family, the PTK focal adhesion kinase (FAK), and several integrin-associated phospho-proteins such as cortactin and tensin (21, 22), also the type I phosphatidylinositol-4-phosphate-5′ kinase gamma (PIP5KIγ) has been implicated in integrin-mediated internalization of *S. aureus* (23). In particular, the 90 kDa integrin-associated isoform of PIP5KIγ (PIP5KIγ90) and its product, phosphatidylinositol-4,5-bisphosphate (PI-4,5-$P_2$), accumulate around *S. aureus* attachment sites (23). Moreover, RNAi-mediated downregulation or full genetic deletion of PIP5KIγ90 reduces the FnBP-mediated, integrin-dependent uptake of the pathogens and only the re-expression of the active enzyme, but not an inactive mutant of PIP5KIγ90 can rescue this phenotype (23). These results demonstrate that elevated levels of PI-4,5-$P_2$ drive *S. aureus* invasion, presumably via the known role of PI-4,5-$P_2$ to serve as a membrane localization signal for cytosolic proteins. For example, several integrin-associated proteins such as FAK, talin, kindlin, vinculin, or α-actinin but also endocytosis-related proteins such as the clathrin adaptor complex AP2 respond with altered localization and/or conformation to the occurrence of PI-4,5-$P_2$ in the inner leaflet of the plasma membrane (24–27). Following internalization, the elevated PI-4,5-$P_2$ levels around *S. aureus* disappear, implying rapid turnover of this membrane lipid (23). In principle, PI-4,5-$P_2$ levels can be regulated via further kinase-mediated phosphorylation of the inositol ring resulting in PI-3,4,5-$P_3$ but also by phospholipase and lipid phosphatase-dependent hydrolysis (28–30). Whether phosphoinositide phosphatases control PI-4,5-$P_2$ levels during the integrin-mediated uptake of *S.aureus* and how they impact this process is currently unknown.

In this study, we focussed on the contribution of human phosphoinositide-5-phosphatases and performed an unbiased functional screen. Here, we report that among the 10 phosphoinositide 5-phosphatases encoded in the mammalian genome (31), synaptojanin1 (SYNJ1) counteracts the accumulation of PI-4,5-$P_2$ levels around cell-associated *Staphylococcus aureus* and limits the uptake of this pathogen. Accordingly, Synj1-deficient cells support enhanced bacterial invasion accompanied by enlarged membrane invaginations. This phenotype is rescued by the re-expression of the wild-type enzyme but not by an inactive lipid phosphatase. Our results underscore the importance of PI-4,5-$P_2$ for bacterial internalization and point to synaptojanin-1 as an enzyme regulating the endocytosis of ligand-bound integrins.

## RESULTS

### A shRNA-based screen identifies Synaptojanin-1 as a negative regulator of *S. aureus* host cell invasion

As we previously reported, the local synthesis of PI-4,5-$P_2$ by PIP5KIγ90 supports *S. aureus* uptake into human cells and the overexpression of an exogenous 5′-directed phosphatidylinositol phosphatase significantly reduces bacterial internalization (23). To investigate, which endogenous phosphoinositide 5-phosphatase(s) regulate integrin-mediated

uptake of *S. aureus*, we targeted individual phosphoinositide 5-phosphatase via RNA interference to generate a panel of lipid phosphatase knockdown cells. These stable knockdown cells were then infected with *S. aureus,* and the amount of cell-associated (total) bacteria as well as viable intracellular bacteria was evaluated by gentamicin protection assays (Fig. S1A and S1B). Compared to cells stably transduced with a control shRNA (Ctrl KD cells), several phosphatase knockdown cells repeatedly showed enhanced numbers of viable intracellular bacteria (Fig. S1B). However, in some cases, the number of total cell-associated bacteria also differed from the Ctrl KD cells indicating that in these cases trafficking and surface localization of the fibronectin-receptor might be influenced (Fig. S1A). When the amount of intracellular bacteria was normalized by the number of total cell-associated bacteria of the corresponding sample, a small set of phosphatases was associated with altered integrin-dependent uptake of staphylococci (Fig. 1A). Especially, in INPP5B, INPP5K, synaptojanin1 (SYNJ1), and SHIP1 knockdown cells, internalization of *S. aureus* was consistently increased to >125% compared to the other phosphatase knockdown or Ctrl KD cells (Fig. 1A). A drawback of antibiotic protection assays, such as the gentamicin assay, relies on the fact that only viable intracellular bacteria are enumerated. As the knockdown of individual phosphoinositide 5-phosphatases might also affect the intracellular trafficking and killing of internalized bacteria, we evaluated the role of these enzymes by an alternative flow cytometer-based assay of bacterial internalization. To this end, *S. aureus* was labeled with fluorescein and, upon infection of INPP5B-, INPP5K-, SYNJ1-, or SHIP1-knockdown cells, the uptake of bacteria was measured by flow cytometry (Fig. 1B). Determination of the fluorescein signal derived from intracellular bacteria showed that depletion of SYNJ1 consistently resulted in increased uptake of *S. aureus*, while depletion of the other candidate phosphatases did not show a major difference compared to control-treated cells (Fig. 1B and C). These results suggested that SYNJ1 is a negative regulator of *S. aureus* internalization via host cell integrins.

## Complementation with active synaptojanin1 reduces the elevated *S. aureus* invasion in synaptojanin1-knock-down cells

Cells stably transduced with the SYNJ1 shRNA showed a ~60% reduction in synaptojanin1 protein levels compared to cells transduced with the control shRNA (Fig. 2A). The control cells and the SYNJ1 knockdown cells were transfected with a GFP encoding construct. Furthermore, the SYNJ1 knockdown cells were transfected with GFP-tagged wild-type synaptojanin1 (GFP-SYNJ1-WT) or an enzymatically inactive mutant of the phosphatase (GFP-SYNJ1-D730A) (Fig. 2B). The complementation of the SYNJ1 knockdown cells with constructs harboring the synaptojanin1 cDNA was possible since the SYNJ1 shRNA-oligonucleotide targeted the 3′ untranslated region of the human synaptojanin mRNA. GFP and the GFP-fusion proteins were expressed at equivalent levels in the transfected SYNJ1 knockdown cells as confirmed by Western blotting (Fig. S1C). To analyze the capability of these cells to internalize *S. aureus*, we used a microscopic approach and differentially stained extracellular and intracellular bacteria in the infected samples (Fig. 2C). Consistent with the initial results, the number of total cell-associated bacteria did not differ between SYNJ1-expressing cells and the SYNJ1 knockdown cells (Fig. 2D). However, *S. aureus* uptake was elevated in SYNJ1 knockdown cells compared to the cells treated with the control shRNA (Fig. 2C and E). Most importantly, re-expression of wild-type synaptojanin1, but not SYNJ1 D730A, in the knockdown cells reduced the number of intracellular bacteria back to the levels found in control cells (Fig. 2C and E). These results indicate that the phosphatase activity of SYNJ1 is critical for its impact on bacterial internalization. To investigate if SYNJ1 acts in the vicinity of bacteria during the internalization process, we expressed GFP and GFP-SYNJ1-WT and infected the cells with the integrin-binding *S. aureus* or the non-pathogenic *S. carnosus*, which does not engage integrins (Fig. 2F). SYNJ1-WT, but not GFP alone, was enriched around *S. aureus* during the internalization process, while contact between non-integrin-binding *S. carnosus* and the cell membrane, which only occurred in rare

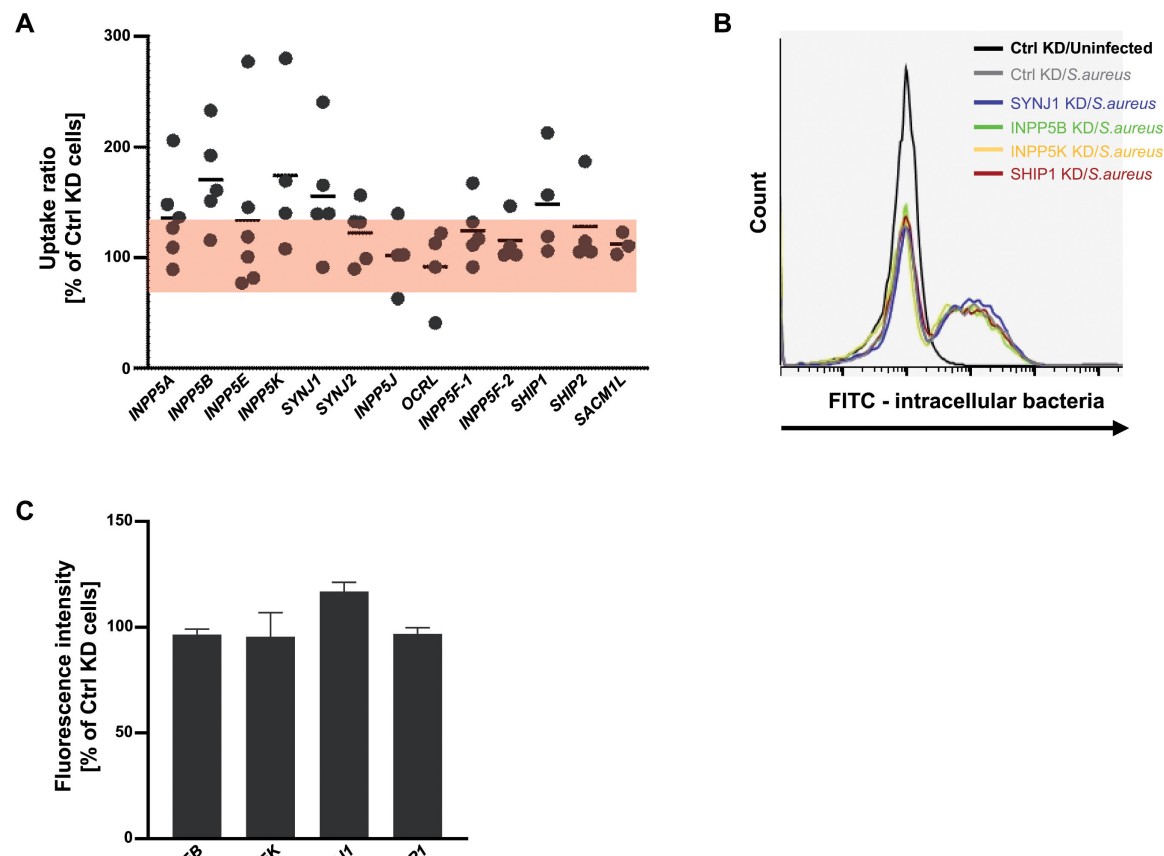

**FIG 1** shRNA-based screen identifies synaptojanin1 as a negative regulator of *S. aureus* host cell invasion. (A) 293 cells with knockdown of individual 5′-phosphatases were infected with *S. aureus* for 2 h. The total cell-associated and the recovered intracellular bacteria were quantified by gentamicin protection assays. The ratio of intracellular vs total cell-associated bacteria for each sample (uptake ratio) was normalized to the uptake ratio of control KD cells, which were set to 100. Each dot represents an independent assay, and horizontal lines indicate mean values (*n* = 4–6). The red bar indicates the range of 75%–125% of the control KD cell uptake ratio. (B) Representative FACS analysis of *S. aureus* internalization in control, SYNJ1, INPP5B, INPP5K, and SHIP1 KD cells. (C) The mean fluorescence intensity in SYNJ1, INPP5B, INPP5K, and SHIP1 KD cells as measured from (B) was normalized by the intensity gained from control KD cells. Bars indicate mean values ± SEM (*n* = 3).

cases, did not result in a redistribution of SYNJ1 (Fig. 2F). In addition to a prominent staining on the membrane surrounding *S. aureus*, wild-type SYNJ1 was also concentrated on vesicles in the vicinity of the bacteria (Fig. 2F). We also tested the localization of the phosphatase-inactive GFP-SYNJ1-D730A and of synaptojanin1 lacking the Sac1 domain (GFP-SYNJ1-ΔSac1). In both cases, these proteins showed a more diffuse distribution in the cytosol of the cell and did not strongly enrich around cell-associated bacteria (Fig. 2G). Together, these findings provide evidence that SYNJ1 co-localizes with *S. aureus* during entry and further demonstrate that the activity of this enzyme limits the integrin-mediated internalization of *S. aureus* into non-professional phagocytes.

## CRISPR/Cas9-mediated knockout of synaptojanin1 facilitates *S. aureus* invasion

Though the shRNA-mediated knockdown leads to reduced SYNJ1 levels, a significant amount of this protein (~40%) remained in the knockdown cells. Therefore, we wondered if residual synaptojanin1 could mask a potentially more severe effect on bacterial internalization. Therefore, we generated clonal Synj1 knockout mouse fibroblasts lines via CRISPR-Cas9 mediated gene disruption. We also treated mouse fibroblasts with control sgRNA to generate a control cell line (Ctrl KO). Western blotting with antibodies

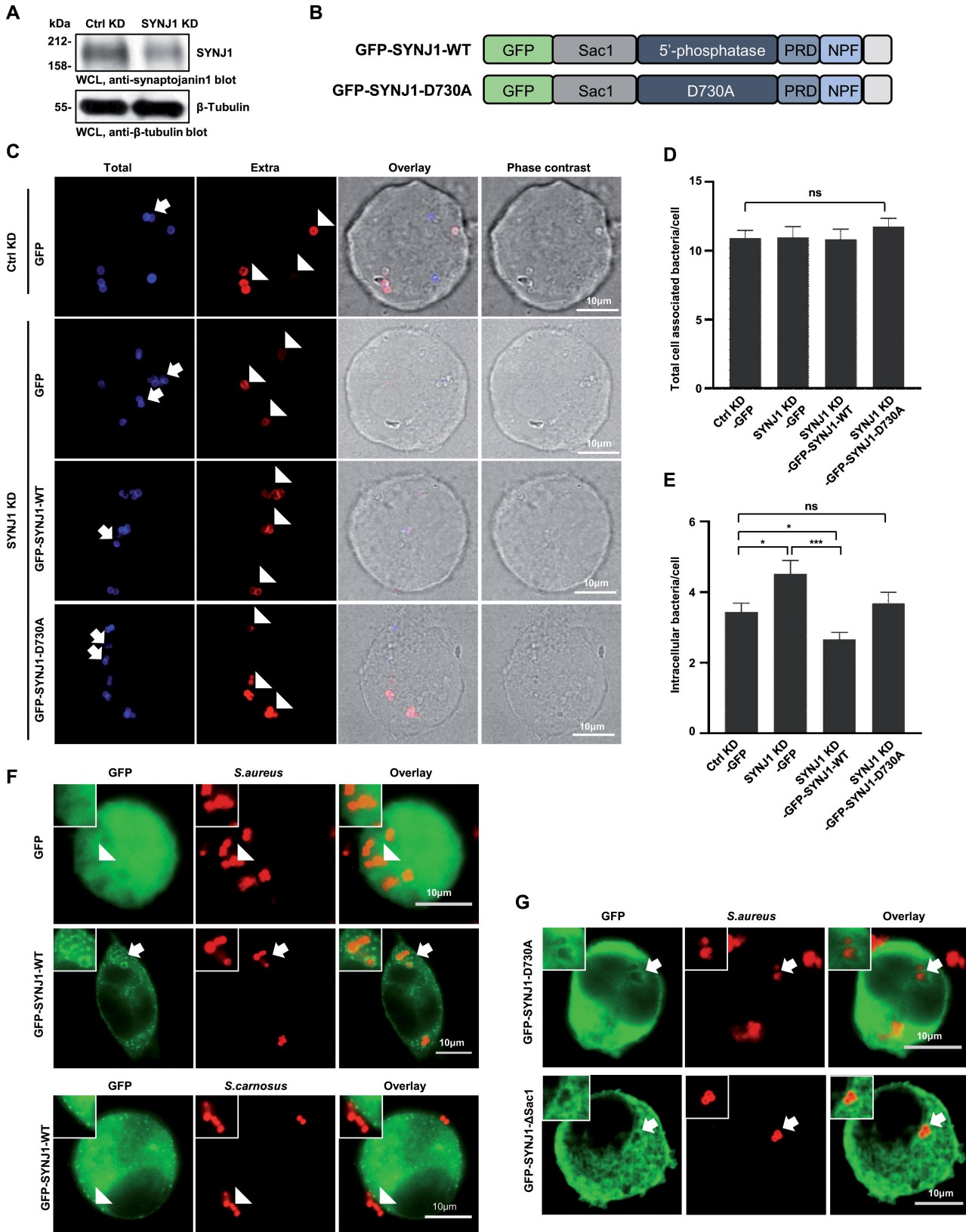

**FIG 2** Complementation with active synaptojanin1 reduces the elevated *S. aureus* invasion in synaptojanin1 knockdown cells. (A) Knockdown efficiency of SYNJ1 in 293 cells was analyzed by Western blot, using antibody against human SYNJ1. (B) Schematic view of GFP-SYNJ1-WT and phosphatase-inactive mutant (Continued on next page)

**FIG 2** (Continued)

GFP-SYNJ1-D730A expression constructs. PRD, proline-rich domain; NPF, Asn-Pro-Phe motif. (C) 293 SYNJ1 KD cells were transfected with plasmids encoding encoding GFP, GFP-SYNJ1-WT, or GFP-SYNJ1-D730A, respectively. Control KD cells were transfected with GFP. Twenty-four hours after transfection, cells were infected with Pacific-Blue stained and biotin-labeled *S. aureus* for 2 h. After fixation, samples were stained with streptavidin-AlexaFluor647 to mark extracellular bacteria. Arrows indicate examples of intracellular bacteria stained in blue only, whereas arrowheads point to examples of extracellular bacteria stained in blue and red. (D and E) Experiments were performed as in (C), and the number of total cell-associated (upper panel) or intracellular bacteria (lower panel) per cell was quantified (at least 30 cells/sample). Bars show the mean values ± SEM from three independent experiments. Significance was evaluated by student's *t* test. ***$P < 0.001$, **$P < 0.01$, *$P < 0.05$. (F) 293 cells were transfected as indicated with GFP or the GFP-tagged wild-type version of human synaptojanin1 (SYNJ1-WT). Transfected cells were infected with TAMRA-SE-labeled *S. aureus* or *S. carnosus*, respectively. Two hours later, the samples were fixed and analyzed by confocal microscopy. The recruitment of GFP-tagged proteins in response to *S. aureus* infection is indicated by arrows, whereas *S. carnosus* did not lead to altered distribution of GFP-SYNJ1 and GFP is not recruited to cell-associated *S. aureus* (arrowheads). Bars, 10 µm. (F) 293 cells were transfected with the GFP-tagged inactive form of SYNJ1 (GFP-SYNJ1-D730A), or a version of synaptojanin1 lacking the amino-terminal Sac1 domain (GFP-SYNJ1-ΔSac1). Cells were infected and analyzed as in (F). Bars, 10 µm.

against murine synaptojanin1 demonstrated that several derived clonal cell lines showed severely diminished or completely absent expression of this protein and we continued with clonal cell line #5 (Synj1-KO cells) (Fig. 3A). The Ctrl KO and the Synj1-KO cells exhibited similar levels of expression of the core focal adhesion proteins (Fig. S2A). Flow cytometry revealed that the Synj1-KO cells had reduced surface levels of integrin α5 and integrin β3, while integrin αv and integrin β1 levels were comparable to Ctrl KO cells (Fig. S2B). Importantly, when the Synj1-KO cells were infected with *S. aureus* and the amount of intracellular bacteria was evaluated with gentamicin protection assays, the number of recovered intracellular bacteria doubled in comparison to the Ctrl KO cells (Fig. 3B). The number of total cell-associated bacteria was slightly, but not significantly reduced in Synj1-KO cells, presumably due to the reduced levels of fibronectin-binding integrins α5β1 and αvβ3 expressed on these cells (Fig. 3B; Fig. S2B). Clearly, the reduced integrin levels did not seem to compromise the ability of Synj1-KO cells to internalize more bacteria. A similar result was obtained when cells were infected with fluorescein-labeled *S. aureus* and the amount of internalized bacteria was evaluated by flow cytometry (Fig. 3C). Again, the Synj1-KO cells showed a strong gain of function with regard to the integrin-mediated uptake of the pathogens (Fig. 3D). These data with Synj1-deficient cells corroborate the results obtained with knockdown cells and demonstrate that synaptojanin1 limits host cell invasion by *S. aureus*.

## Synaptojanin1 deficiency leads to exaggerated PI-4,5-P$_2$ accumulation around cell-associated *S. aureus*

Synaptojanin1 activity is directed toward PI-4,5-P$_2$, and this phosphoinositide seems to play a positive role in *S. aureus* uptake (23). Therefore, we speculated that the lack of Synj1 might promote elevated levels of PI-4,5-P$_2$ at bacterial attachment sites leading to increased endocytosis of the bacteria. To examine the local abundance of PI-4,5-P$_2$ upon *S. aureus* infection, Synj1-KO cells and control KO cells were transfected with vectors encoding the GFP-tagged PLCδ-PH domain as PI-4,5-P$_2$-specific probe or GFP alone. Transfected cells were infected with rhodamine-labeled *S. aureus* or *S. carnosus* for 2 h, before the samples were analyzed by confocal microscopy (Fig. 4A). As seen before, GFP did not alter its distribution within the cells upon bacterial infection, whereas PLCδ PH-GFP was recruited to *S. aureus* attachment sites in both control and Synj1-KO cells (Fig. 4A). However, recruitment of the PI-4,5-P$_2$-binding PLCδ-PH domain was clearly enhanced in the vicinity of *S. aureus* attaching to the Synj1-KO cells compared to the CTRL KO cells (Fig. 4A). The increased recruitment of PLCδ PH-GFP was also evident when we quantitatively evaluated the GFP enrichment ratio as the ratio between the GFP-signal intensity in the vicinity of the bacteria vs the overall GFP signal (Fig. 4B). The strong enrichment of PLCδ PH-GFP in the Synj1-KO cells was observed consistently, while only on rare occassions *S. aureus* attachment induced a prominent recruitment of PLCδ PH-GFP in the control KO cells (Fig. 4C). These data demonstrate that the increased uptake of *S. aureus* by Synj1-deficient cells is correlated with local enrichment of PI-4,5-P$_2$ at

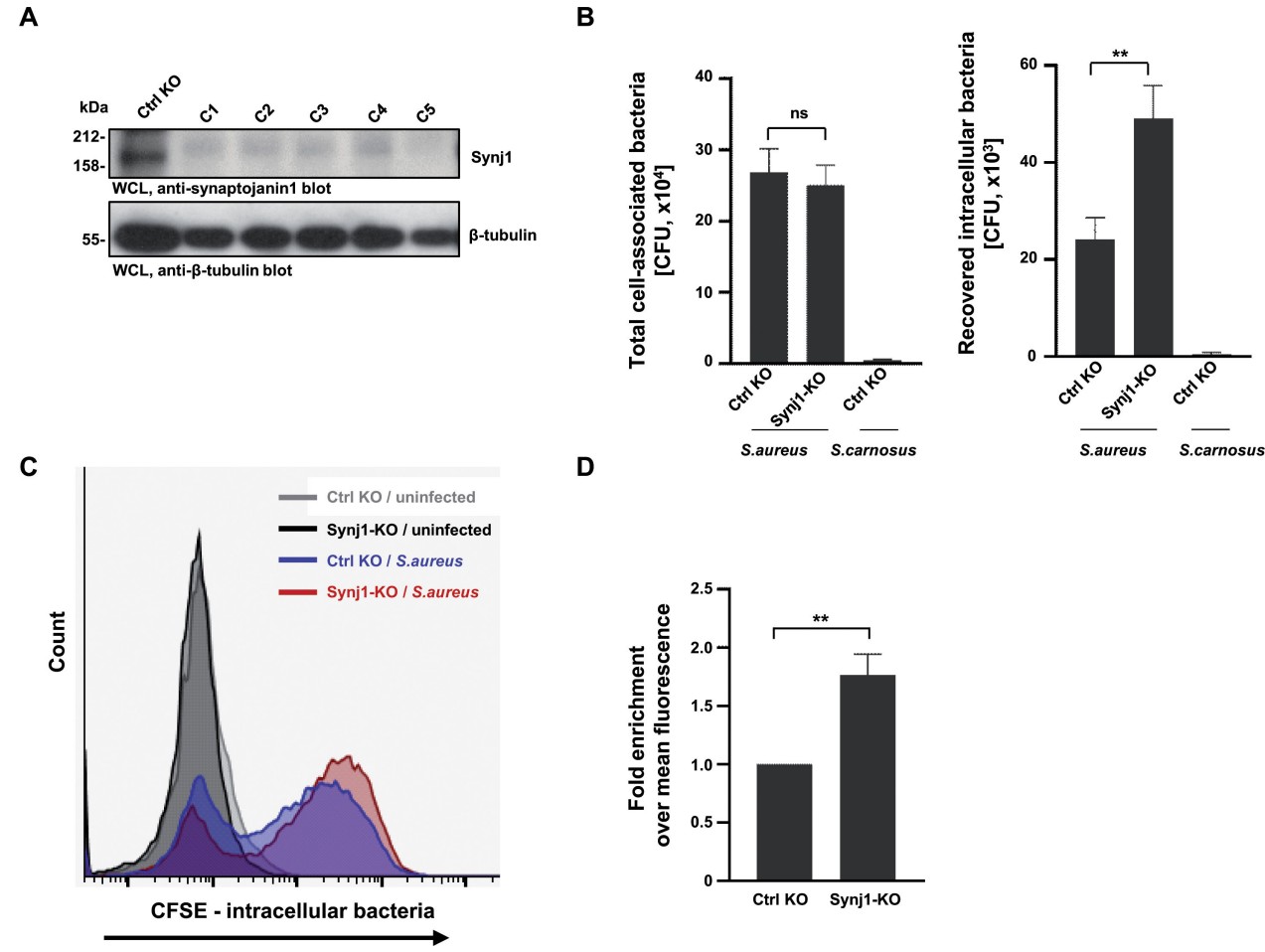

**FIG 3** CRISPR-Cas9 mediated knockout of synaptojanin1 facilitates *S. aureus* invasion. (**A**) Knockout efficiency of Synj1 in NIH3T3 cells was verified by Western blot using antibody against mouse synaptojanin1. (**B**) Control and Synj1-KO cells were infected with *S. aureus* or *S. carnosus*, respectively, for 2 h. The total cell-associated and recovered intracellular bacteria were quantified by gentamicin protection assay. The bars show mean values ± SEM of three independent experiments. Significance was evaluated by an unpaired *t* test. \*\*$P < 0.01$. (**C**) Control and Synj1-KO cells were infected with CFSE-stained *S. aureus* for 2 h. Afterward, cell-associated fluorescein intensity was measured by flow cytometry in the presence of trypan blue. A representative experiment is shown. (**D**) Quantification of bacterial uptake by control and Synj1-KO cells as in (C). The fluorescein intensity of infected control KO cells was set to 1, and the value from Synj1-KO cells was normalized to the value from control KO cells. Bars represent mean FITC intensity of three independent experiments. Significance was evaluated by unpaired *t*-test. \*\*$P < 0.01$.

bacterial attachment sites further supporting the idea that internalization of bacteria is facilitated by this membrane lipid.

## The Sac1-domain of synaptojanin1 is involved in limiting *S. aureus* invasion

Synaptojanin1 is special among 5′-phosphatases as it has a second enzymatic activity encoded in the amino-terminal Sac1 domain (32). This domain mediates dephosphorylation of additional phosphatidylinositol-phosphates, mainly phosphatidylinositol 4-phosphate (PI-4-P) (32). To address the role of this activity in the integrin-mediated uptake of *S. aureus*, we stably re-expressed GFP, a GFP-fused wild-type SYNJ1 (GFP-SYNJ1-WT), or a Sac1 domain-deleted mutant of Synaptojanin-1 (GFP-SYNJ1-ΔSac1) in Synj1-KO cells (Fig. 5A). Compared to the Synj1-KO cells, the SYNJ1-re-expressing Synj1-KO cells showed a protein band at ~200 kDa in line with the expected size of a GFP-SYNJ1 fusion protein (Fig. 5A and B). The GFP-SYNJ1-ΔSac1 mutant was strongly expressed in the Synj1-KO cells and migrated at about 160 kDa (Fig. 5A and B). When employed in gentamicin protection assays, the stable re-expression of wild-type SYNJ1

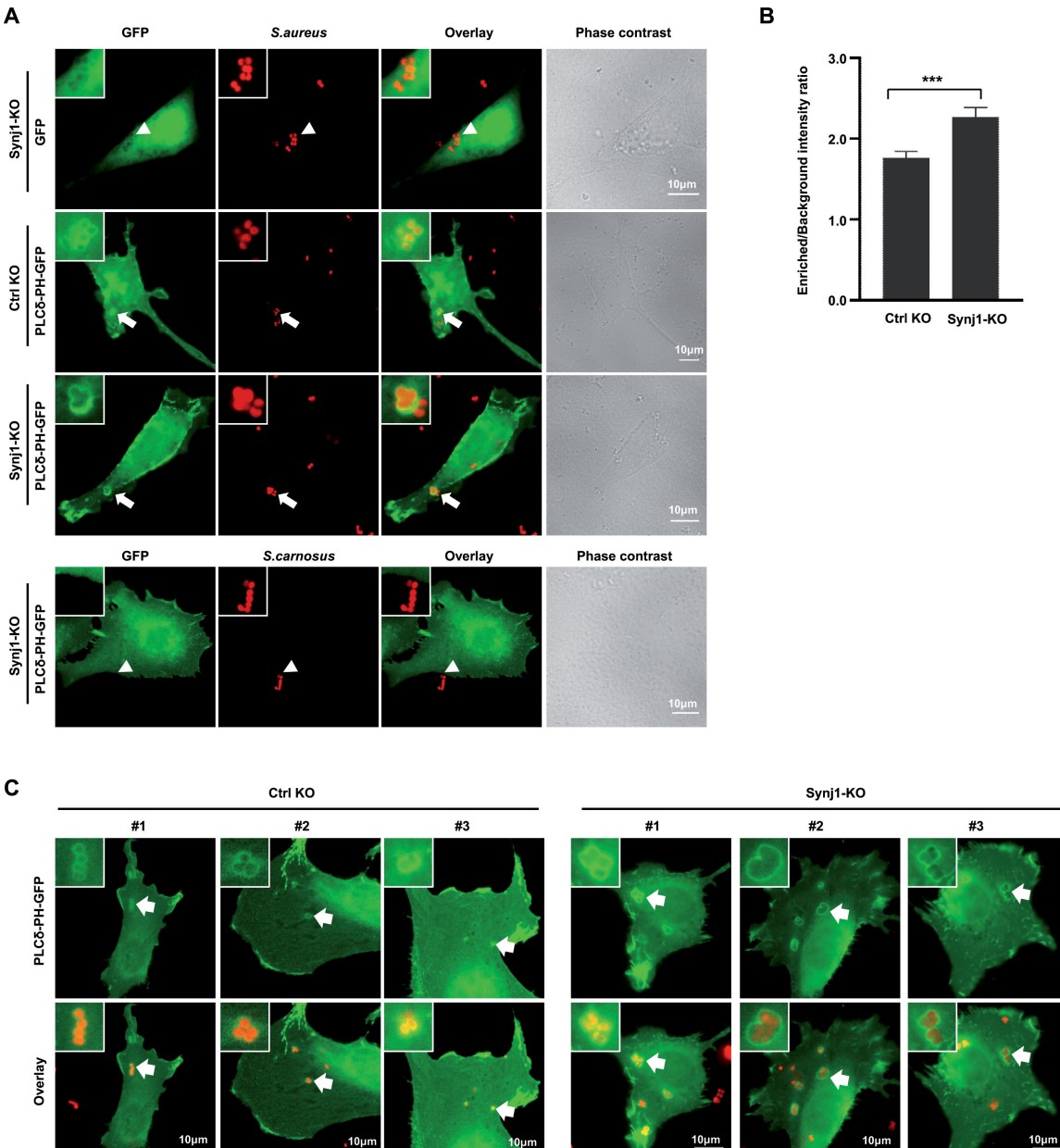

**FIG 4** Synaptojanin1 deficiency leads to exaggerated PI-4,5-P$_2$ accumulation around cell-associated *S. aureus*. (**A**) Control KO and Synj1-KO cells were transfected with vectors encoding GFP-tagged PLCδ-PH domain or GFP alone. Twenty-four hours after transfection, cells were infected with TAMRA-labeled *S. aureus* or *S. carnosus* at MOI 30 for 2 h. Upon fixation, recruitment of GFP or PLCδ-PH-GFP was monitored by confocal fluorescence microscopy. Recruitment of PLCδ-PH-GFP in cells infected with *S. aureus* is indicated by arrows, whereas the absence of GFP enrichment or the lack of PLCδ-PH-GFP recruitment to the rare *S. carnosus*–host cell attachment sites is marked by arrowheads. Bars, 10 µm (B) Quantification of GFP fluorescence intensity in samples from (A). The maximum GFP intensity at the infection sites was divided by the mean fluorescence intensity of the whole cell. The fold GFP enrichment from at least 30 cells for each group is analyzed by GraphPad Prism 7. Significance was evaluated by unpaired *t*-test. \*\*\**P* < 0.001. (C) Representative images of PLCδ-PH-GFP enrichment in control KO and Synj1-KO cells infected with *S. aureus*. Bars, 10 µm.

again reduced the heightened bacterial internalization observed in the GFP only expressing Synj1-KO cells (Fig. 5C). Surprisingly, even the strong expression of GFP-SYNJ1-ΔSac1 was not able to alter the elevated bacterial internalization, suggesting that a functional Sac1 domain is also critical for proper function of synaptojanin1 during *S. aureus* invasion (Fig. 5C). The lack of phenotypic reversion of the Synj1-KO cells by the

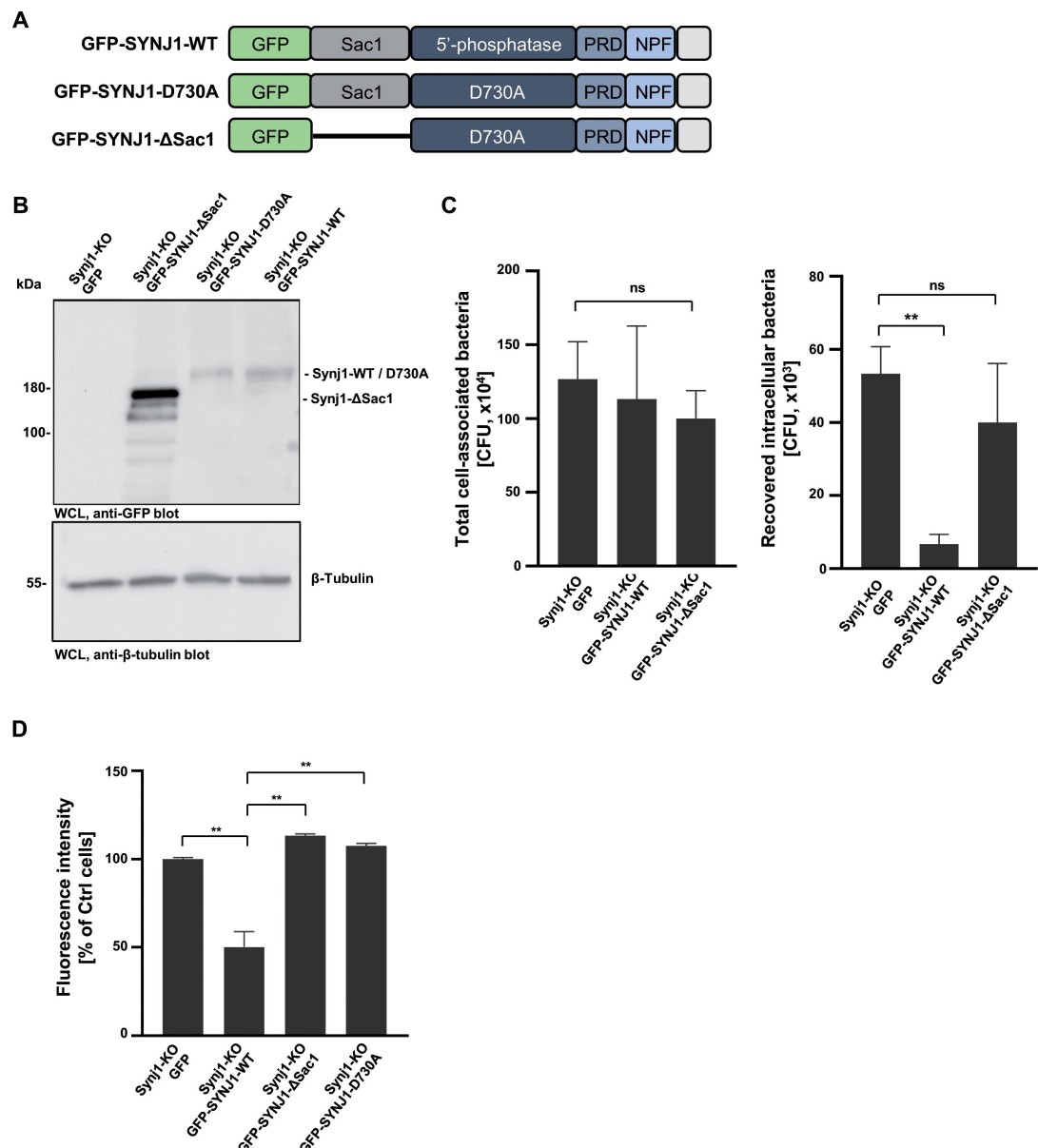

**FIG 5** The Sac1-domain of synaptojanin1 is involved in limiting *S. aureus* invasion. (A) Schematic view of GFP-SYNJ1 WT, GFP-SYNJ1 D730A (5′-phosphatase dead), and GFP-SYNJ1-ΔSac1 (Sac1-domain deleted) expression constructs. PRD, proline-rich domain; NPF, Asn-Pro-Phe motif. (B) The stable re-expression of GFP-SYNJ1 WT and SYNJ1 mutants was verified by anti-GFP Western blot. (C) Synj1-KO cells with stable re-expression of GFP-SYNJ1-WT, GFP-SYNJ1-ΔSac1, or GFP alone were infected with *S. aureus* for 2 h. The total cell-associated and recovered intracellular bacteria were quantified by gentamicin protection assay. The bars show mean values ± SEM of at least two independent experiments done in triplicate. Significance was evaluated by unpaired *t*-test. \*\**P* < 0.01. (D) Synj1-KO cells with stable re-expression of GFP-SYNJ1-WT, GFP-SYNJ1-ΔSac1, GFP-SYNJ1 D730A, or GFP alone were infected with FITC-labeled *S. aureus* for 2 h. Bacterial uptake was determined by flow cytometry. Background GFP fluorescence of uninfected cells was subtracted. Bars indicate mean FITC intensity of internalized bacteria ± SEM (*n* = 4). Significance was evaluated by two-tailed *t*-test. \*\**P* < 0.01.

expression of the SYNJ1-ΔSac1 mutant resembled the results obtained with the re-expression of the 5-phosphatase inactive mutant SYNJ1 D730A in SYNJ1 knockdown cells. To directly compare these two mutants, we also established Synj1-KO cells stably re-expressing the 5-phosphatase inactive mutant (GFP-SYNJ1-D730A) (Fig. 5B). We also confirmed that expression of integrin β1 was unaltered in the Synj1-KO cells upon expression of either GFP-fusion protein (Fig. S2D). Importantly, when bacterial internalization was analyzed by flow cytometry, the wild-type SYNJ1-re-expressing cells showed

reduced numbers of internalized bacteria when compared to the Synj1 knockout cells (Fig. 5D). However, both the 5-phosphatase inactive mutant GFP-SYNJ1-D730A and the GFP-SYNJ1-ΔSac1 enzyme were not able to revert the elevated uptake of *S. aureus* and even harbored slightly increased numbers of intracellular bacteria than the Synj1-KO cells (Fig. 5D). These complementation experiments of the Synj1 knockout cells confirmed that synaptojanin1 is a negative regulator of integrin-mediated internalization of *S. aureus* and that both enzymatic activities of this lipid phosphatase are necessary in this context.

## Lack of synaptojanin1 increases talin recruitment and leads to enlarged invaginations during *S. aureus* uptake

One of the cytoplasmic factors, which is regulated by PI-4,5-$P_2$ binding and which is critical for integrin activation, is the scaffolding protein talin (33, 34). The intramolecular autoinhibition of talin is relieved by PI-4,5-$P_2$ binding (35), and talin recruitment to cell-associated *S. aureus* is strongly impaired in PIP5KI$\gamma$90$^{-/-}$ cells (23). Based on these findings, we wondered whether the increased PI-4,5-$P_2$ levels in Synj1-KO cells might lead to altered talin recruitment upon *S. aureus* infection. Therefore, Ctrl KO and Synj1-KO cells were transfected with a plasmid encoding GFP-tagged talin or a plasmid encoding GFP alone. Next, the transfected cells were infected with rhodamine-stained *S. aureus* or *S. carnosus* for 2 h, and the fixed samples were analyzed by confocal scanning microscopy. While GFP was not enriched around *S. aureus*, GFP-talin showed weak recruitment to the bacteria in Ctrl KO cells (Fig. 6A). However, in Synj1-KO cells, a strong enrichment of GFP-talin around the cell-associated pathogens was observed (Fig. 6A). The distribution of talin was not influenced by the occasional contact of non-pathogenic *S. carnosus* with Synj1-KO cells (Fig. 6B). Quantification of GFP-intensity through bacterial attachment sites indicated a ~30% increase of GFP-talin recruitment in Synj1-KO cells compared to Ctrl KO cells (Fig. 6C). To see, if the enrichment of talin and PI-4,5-$P_2$ altered the cellular morphology during bacterial internalization, we analyzed infected samples by scanning electron microscopy. Synj1-KO cells or the different stable re-expressing cell lines were infected for 1 h with *S. aureus* to observe the initial formation of uptake sites. Indeed, Synj1-KO cells showed numerous individual and sometimes greatly enlarged uptake sites, while the GFP-SYNJ1 wild-type re-expressing cells did hardly internalize bacteria (Fig. 6D). Strikingly, in both the GFP-SYNJ1-D730A and the GFP-SYNJ1-ΔSac1 cells, the exaggerated surface invaginations containing numerous *S. aureus* were even more prominent than in the Synj1-KO cells (Fig. 6D). The presence of oversize uptake sites in Synj1-KO cells and in cells re-expressing the enzymatically inactive versions of this lipid phosphatase implies that the absence of functional synaptojanin1 has an impact on endosome morphology. The elevated PI-4,5-$P_2$ levels found in these cells appear to promote invagination formation helping to explain the increase in integrin-mediated uptake of *S. aureus*.

## DISCUSSION

PI-4,5-$P_2$ embedded in the inner leaflet of the eukaryotic plasma membrane participates in several critical cellular processes. This negatively charged lipid is connected to membrane deformation and endocytosis but is also known to orchestrate the assembly of integrin-associated focal adhesion complexes at the cell membrane (26, 27, 36). While the generation of PI-4,5-$P_2$ by a focal adhesion localized PI-5 kinase has been studied in detail, it is currently unknown, which PIP-directed phosphatase regulates integrin-initiated, PI-4,5-$P_2$-dependent processes. Here, we report that the ubiquitously expressed PI-4,5-$P_2$-directed lipid phosphatase synaptojanin1 counteracts the internalization of integrin-binding *Staphylococcus aureus*. Indeed, deficiency of synaptojanin1 leads to a gain of function with regard to *S. aureus* invasion into different non-phagocytic cell types accompanied by increased local levels of PI-4,5-$P_2$ and elevated recruitment of the PI-4,5-$P_2$ -binding integrin activator talin. These findings suggest that the lipid

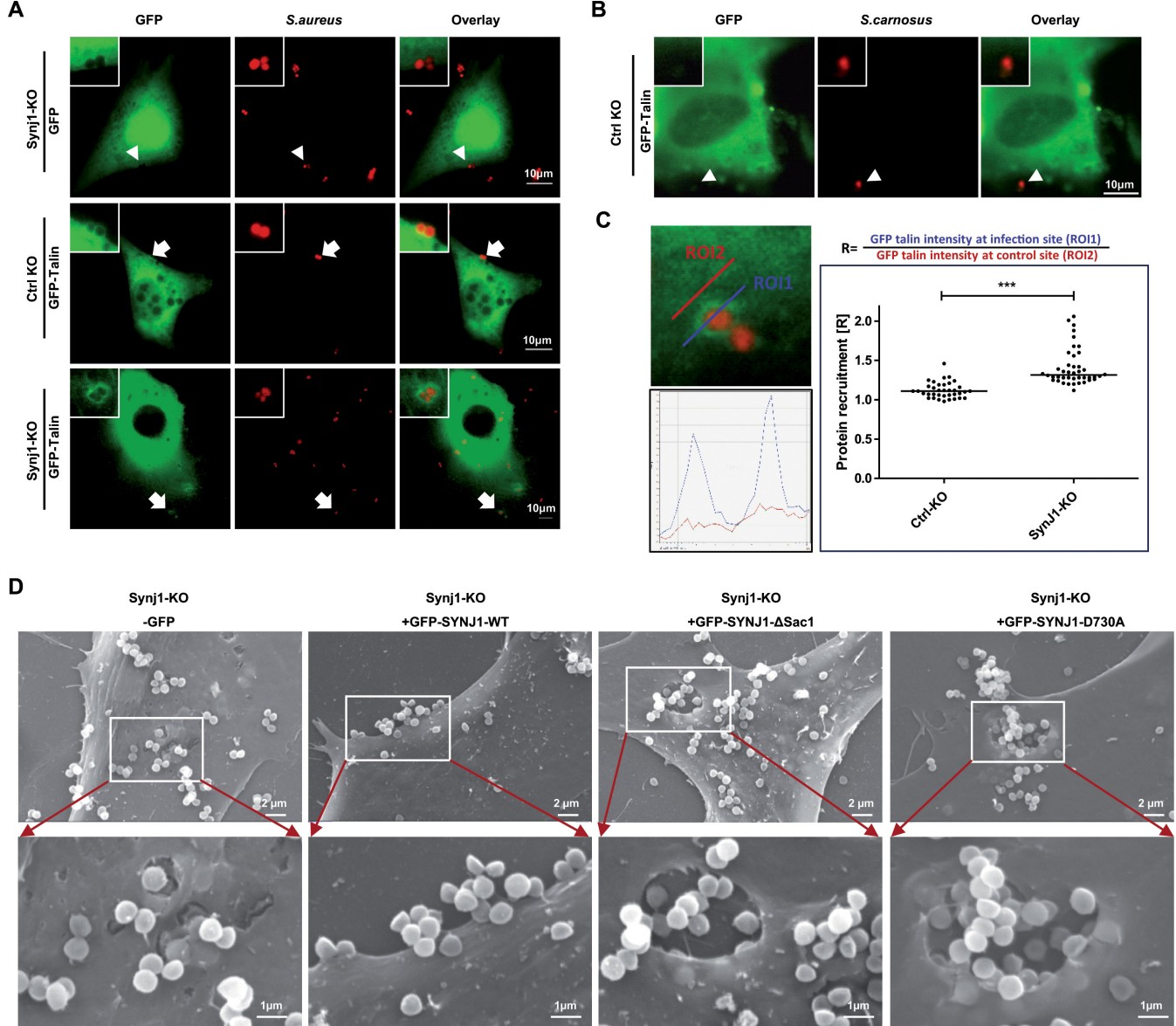

**FIG 6** Lack of synaptojanin1 increases talin recruitment and leads to enlarged invaginations during *S. aureus* uptake. (A) Control KO or Synj1-KO cells were seeded on poly-lysine coated coverslips in 24-well plates and transfected with vector encoding GFP-tagged talin or GFP alone. Forty-eight hours after transfection, cells were infected with TAMRA-labeled *S. aureus* for 2 h. After fixation, samples were examined by fluorescence microscopy. The recruitment of GFP-talin is indicated by arrows. GFP is not recruited to bacteria (arrowheads). Scale bars, 10 µm. (B) Control KO cells were transfected with GFP-tagged talin, infected with TAMRA-labeled *S.carnosus* for 2 h and evaluated as in (A). (C) In samples of *S. aureus*-infected Control KO or Synj1-KO cells from (A), recruitment of GFP-tagged talin was quantified along line scans through individual bacterial attachment sites (ROI1; blue) and neighboring control sites (ROI2; red) as indicated in the left panels. The graph depicts the GFP-talin recruitment ratio at individual bacterial attachment sites, line indicates median value of the two samples ($n$ = 36 and 42, respectively). Significance was evaluated by Mann-Whitney $U$ test. ***$P$ < 0.001. (D) Synj1-KO cells stably expressing GFP, GFP-SYNJ1-WT, GFP-SYNJ1-D730A, or GFP-SYNJ1-ΔSac1 were infected with *S. aureus* for 2 h. Fixed samples were analyzed by scanning electron microscopy. Shown are representative images. Scale bars, 2 µm (upper panel), 1 µm (lower panel).

phosphatase synaptojanin1 not only limits the integrin-mediated uptake of pathogenic bacteria but might also regulate integrin-dependent processes in different cell types under physiological conditions.

While the complete engulfment of enteropathogenic *Yersinia pseudotuberculosis* seems to require PI-4,5-P$_2$ hydrolysis by inositol 5-phosphatases OCRL and INPP5B (37), our study adds to a growing body of evidence suggesting that the presence

of PI-4,5-P$_2$ promotes bacterial uptake by non-phagocytic cells. Indeed, this has been observed in the case of *Listeria monocytogenes*, which triggers internalization upon binding of the host receptor c-Met (38). Engagement of c-Met by the bacterial surface protein InlB induces the recruitment of OCRL to *L. monocytogenes* invasion foci (38). Similar to our results with synaptojanin-1 depletion during integrin-triggered uptake of *S. aureus*, the siRNA-mediated knockdown of OCRL results in enhanced *Listeria* internalization by human cells. These findings highlight the negative impact of inositol 5-phosphatase-mediated PI-4,5-P$_2$ hydrolysis on bacterial invasion. Moreover, in both cases, the elevated PI-4,5-P$_2$ levels in the absence of the inositol 5-phosphatase are accompanied by reduced surface levels of the involved host receptors. This phenomenon could be due to an altered endocytic recycling of receptors, which might result as a consequence of higher PI-4,5-P$_2$ levels in the plasma and, potentially, endosomal membranes (39). However, despite reduced c-Met levels or reduced integrin α5 levels on OCRL- or SYNJ1-deficient cells, respectively, the uptake of receptor-bound pathogens is consistently increased in both cases. Therefore, an altered endosomal recycling of receptors, which appears to be connected to reduced receptor levels available on the membrane surface, would not explain the elevated uptake of the bacteria.

In the case of *S. aureus*, which exploits the plasma protein fibronectin to engage integrins, the multivalent pathogen seems to cluster integrins and integrin-associated proteins to trigger the assembly of focal adhesion- and fibrillar adhesion-like protein complexes (16, 21–23). It is interesting to note that focal adhesions are enriched in proteins possessing PI-4,5-P$_2$-binding capabilities. In particular, the main activators of integrin-binding function, proteins of the talin and kindlin families, require PI-4,5-P$_2$ for their recruitment and conformational regulation. Both integrin activators are characterized by amino-terminal FERM-domains having PI-4,5-P$_2$ binding pockets, which are involved in orchestrating intramolecular interactions (40, 41). Furthermore, additional integrin-associated scaffolding and signaling proteins such as vinculin, α-actinin, and focal adhesion kinase (FAK) are modulated by PI-4,5-P$_2$ binding (42–44). It is, therefore, not surprising that a specific isoform of type I phosphatidylinositol-4-phosphate-5′ kinase gamma, PIP5KIγ90, localizes to focal adhesions by associating with talin, where it contributes to the local generation of PI-4,5-P$_2$ (45–47). Mice engineered to lack this particular isoform are viable and fertile, but PIP5KIγ90-deficient cells exhibit an adhesion defect associated with reduced recruitment of talin and vinculin to focal adhesion sites (48, 49). Interestingly, this phenotype is mirrored by our results, where the downregulation of synpatojanin1 results in increased recruitment of talin. Moreover, depletion of PIP5KIγ90 and depletion of synaptojanin1 result in opposing phenotypes with regard to the integrin-mediated internalization of *S. aureus* (23). These findings indicate that synaptojanin1 is the cellular 5′-phosphatase, which counteracts the local activity of PIP5KIγ90 in the context of integrin-mediated endocytosis.

As observed before, the reduced local production of PI-4,5-P$_2$ in fibroblasts and platelets lacking the integrin-associated isoform PIP5KIγ90 does not seem to diminish integrin-ligand binding (48, 50). In agreement with these findings, we also observe comparable binding of fibronectin-associated stapylococci to the wildtype as well as to synaptojanin1-knockout cells. However, in the absence of local PI-4,5-P$_2$, a reduced force coupling between membrane-embedded integrins and the intracellular cytoskeleton has been observed, which might result from the reduced recruitment of actin-associated and PIP2-binding proteins to FA sites under these conditions (48, 51). Along these lines of thought, compromising the local turnover of PI-4,5-P$_2$ by targeting the responsible lipid phosphatase should strengthen the link between membrane receptors and the cytoskeleton. Indeed, the increased magnitude and extent of membrane invaginations seen by scanning electron microscopy in synaptojanin1-deficient cells points to a strengthened mechanical linkage between bacteria-engaged integrins and the cytoskeleton. This improved force transmission could translate into enhanced membrane bending around integrin-associated bacteria, finally leading to the elevated numbers of intracellular staphylococci.

It is interesting to note that besides the 5′-phosphatase activity, also the Sac1 domain of synaptojanin1, which by itself can exhibit 4′-phosphatase activity, was required to completely revert the gain of function seen in SYNJ1-deficient cells. How these different enzymatic activities of synaptojanin combine to regulate endocytosis of ligand-occupied integrins will be an important research topic to understand the role of this protein beyond bacterial internalization in different physiological contexts.

## MATERIALS AND METHODS

### Bacteria

*E. coli* Nova Blue was cultured in Lysogeny Broth (LB) medium at 37°C. *S. aureus* Cowan and non-pathogenic *S. carnosus* TM300 were cultured in Tryptic Soybean Broth medium (TSB; BD Biosciences, Heidelberg, Germany) at 37°C. To infect host cells, staphylococci were harvested at exponential growth phase at ~0.6 $OD_{600}$. Bacteria were washed twice with PBS and then added to the cells at a multiplicity of infection (MOI) 20 for gentamicin protection assay/FACS assay or MOI 30 for microscopic evaluation. In certain assays, PBS-washed staphylococci from log-phase cultures were labeled with 1 µg/mL 5-(6)-carboxy-fluorescein-succinimidylester (CFSE), 5-(and-6)-carboxy-tetramethyl-rhoda-mine-succinimidylester (TAMRA-SE), Pacific Blue-succinimidylester (PBSE), or Sulfo-NHS-LC-biotin (all purchased from ThermoFisher, Life Technologies, Darmstadt, Germany) for 20 min at room temperature prior to infection as previously described (14, 52, 53).

### Cell culture and transfection

Human embryonic kidney 293T cells (293 cells; American Type Culture Collection CRL-3216) were cultured in DMEM supplemented with 10% calf serum (CS) at 37°C and 5% $CO_2$. NIH 3T3 Flp-In cells (Invitrogen) were cultured in DMEM with 10% FCS. All cell lines were sub-cultured every 2–3 days and were regularly checked by PCR according to reference (54) for the absence of mycoplasm contaminations. For transfection of HEK293T cells, standard calcium phosphate co-precipitation with 1–5 µg of plasmid DNA for each 10 cm culture dish was used. NIH 3T3 Flp-In cells were transfected using Lipofectamine 2000 reagent (Invitrogen, Carlsbad, CA) according to the manufacturer's instructions. Cells were employed in infection experiments 24–48 h after transfection.

### DNA constructs

The expression construct encoding PLCδ-PH-GFP was provided by Tamás Balla (NIH, Bethesda, MD) and pRK GFP-Talin was provided by Reinhard Fässler (MPI for Bio-chemistry, Martinsried, Germany). pEGFP-C1-hSYNJ1-170 wt and the hSYNJ1 D730A mutant were generously provided by Pietro De Camilli (Yale University School of Medicine, New Haven). The cDNA of human SYNJ1-WT and SYNJ1-ΔSac1 were cloned into vector pDNR-dual-LIC (55) via LIC cloning using pEGFP-C1-hSYNJ1-170 as tem-plate. For SYNJ1-WT, the following primers were used: for_5′-ACTCCTCCCCCGCCATG GCGTTCAGCAAAGGATTTCG-3′ and rev_5′-CCCCACTAACCCGTTATCTTTCTGTAAAGTCCAG -3′. For SYNJ1-ΔSac1, the following primers were used: for_5′-ACTCCTCCCCCGCCATGG GAACTGGAGCTCTTGAAG-3′ and rev_5′-CCCCACTAACCCGTTATCTTTCTGTAAAGTCCAG-3′. SYNJ1-D730A encoding cDNA was inserted into pDNR-dual-LIC in a similar manner using pEGFP-C1-hSYNJ1-170 D730A as template with following primers: for_5′-ACTCC TCCCCCGCCATGGCGTTCAGCAAAGGATTTCG-3′ and rev_5′-CCCCACTAACCCGTTATCTTTCT GTAAAGTCCAG-3′. The cloned cDNAs were transferred by Cre-mediated recombination from pDNR-dual-LIC into the LoxP site of plasmid pEF5-FRT-eGFP-C1. pEF5-FRT-eGFP-C1 was constructed by inserting the SpeI/PspXI fragment containing the eGFP cDNA and a C-terminal, in-frame LoxP site from vector pEGFP-C1-LoxP (56) into the SpeI/ PspXI restricted vector pEF5/FRT-DEST EVC2ala4-Flag (gift from Rajat Rohatgi; Addgene plasmid #41008).

The shRNAs used for 5′-phosphatase knockdown were annealed in the form of double-stranded DNA oligos with corresponding 5′-overhangs to insert them into the AgeI/EcoRI-restricted vector pLKO.1-TRC (a gift from David Root) (Addgene plasmid #10878). The used sequences of shRNAs were as follows: hINPP5A, AAGTCACAGTCCTG TTGTCAA; hINPP5B, AAACTCCTGAACTCAGGTAAT; hINPP5E, AATCTAGGGACATTCAATCTA; hINPP5K, AATTAGCCGCTTAAATACAGG; hSYNJ1, AAACCCTTCTCATTGTTAACT; hSYNJ2, AA TCGTGTCTCTTATTCAGTA; hINPP5J, AATCCTGGAGGTCATCCATTA; hOCRL, AAAGGAAGAGC GTTTCCTAAT; hINPP5F-1 (NM_001243194.1 and NM_014937.3), AAACCTAGGAGCATTAAG ACT; hINPP5F-2 (NM_001243195.1), AACAAACCAGAGAAGATCATA; hSHIP1, AAGCCATTCT GAAGAAAGGAA; hSHIP2, AAGCAGTATCTCTCTGCCTAT; hSACM1L, AAGGACCAACTTAAAC GTTAA.

For the generation of sgRNA encoding plasmids, the U6 promoter was amplified from plasmid pSpCas9(BB)-2A-Puro (PX459) (a gift from Feng Zhang; Addgene plasmid #48139) with primers U6_gRNA_KpnI_for 5′-ATAGGTACCGTGAGGGCCTATTTCCC and U6_gRNA_XhoI_rev 5′-ATACTCGAGCTATTTGTACAGTTCGTCCATGCCG, and the resulting PCR fragment was inserted into the KpnI/XhoI sites of pBluescript SK+ (Stratagene). Upon cleavage with BbsI, annealed sgRNA oligos targeting the murine SYNJ1 gene were inserted. Plasmid pBS-U6 sgRNA SYNJ1-exon1 resulted from annealed oligos 5′- C ACCGTAAGGGGCGGGCCTTCCGCC-3 and 5′-AAACGGCGGAAGGCCCGCCCCTTAC-3′, while plasmid pBS-U6 sgRNA SYNJ1- exon2 resulted from annealed oligos 5′-CACCGAGGAGAA TGGCGTTCAGCAA-3′ and 5′-AAACTTGCTGAACGCCATTCTCCTC-3′.

## Lentivirus production and generation of stable cell lines

The lentiviral vectors pLKO.1-TRC (#10878), pMD2.G (#12259), and psPAX2 (#12260) were purchased from Addgene. HEK293T cells were transfected by standard calcium-phosphate co-precipitation, with 7 µg pMD2.G, 10 µg psPAX2 and 13 µg pLKO.1 harboring the corresponding shRNA. Seventy-two hours after transfection, the virus-containing supernatant was collected, cleared by centrifugation for 7 min at 2,000 rpm, and sterile-filtrated. HEK293T cells were directly incubated with the corresponding lentiviral supernatant for 24 h and afterward selected with fresh cell culture medium containing 0.4 µg/mL puromycin. Control cells were generated via transduction with virus harboring the empty pLKO.1.

## Gentamicin protection assay

$2 \times 10^5$ 293 cells or $5 \times 10^4$ NIH 3T3 cells were seeded into poly-L-lysine coated 24-well plates. Cells were infected at MOI 20 for 2 h at 37°C and 5% $CO_2$. Two sets of triplicate wells with identically infected cells were processed in two different ways: (i) to evaluate the number of viable intracellular bacteria, the medium was carefully replaced with a fresh medium containing 50 µg/mL gentamicin. After incubation for 1 h at 37°C, intracellular bacteria were released by incubation with 0.5% saponin for 15 min at 37°C. Released bacteria were diluted in PBS and plated on TSB agar plates to determine the colony-forming units (cfu). Colony counts of these samples are referred to as "recovered intracellular bacteria." (ii) to determine the total number of cell-associated intracellular and extracellular bacteria, the infected cells were gently washed with PBS and lysed with 0.5% saponin without prior gentamicin treatment, and dilutions were plated on TSB agar plates. Colony counts of these samples are referred to as "total cell-associated bacteria."

## Evaluation of bacterial internalization by flow cytometry

For flow cytometric analysis of bacterial uptake, cells were infected with CFSE-labeled bacteria and analyzed as described (53). Briefly, $4 \times 10^5$ HEK293 cells or $1 \times 10^5$ fibroblasts were seeded 1 day before infection in poly-L-lysine or gelatine-coated 6-well plates. PBS-washed bacteria from logarithmic growing cultures were stained with 5-(6)-carboxyfluorescein-succinimidylester (CFSE) and added at a multiplicity of infection (MOI) of 20. After 2 h, infected cells were detached and washed with PBS, and cell-associated

fluorescein fluorescence was analyzed by flow cytometry. To quench signals from extracellular bacteria, trypan blue solution (0.4%; Sigma, Taufkirchen, Germany) was added to a final concentration of 0.2% directly before analysis. Samples were analyzed on a LSRII (BD Biosciences) by gating on the eukaryotic cells based on forward and side scatter and cell-associated fluorescence of 10,000 cells per sample was measured in fluorescence channel 1 (FL1-H) detecting CFSE fluorescence as a measure of internalized bacteria.

## Extra- and intracellular bacteria staining and fluorescence microscopy examination

For the experiment performed in 293 cells, $2 \times 10^5$ transfected cells were seeded on poly-L-lysine coated glass coverslips in a 24-well plate. Two hours later, cells were infected with Pacific-Blue- and biotin-labeled *S. aureus* at MOI 30 for 2 h, washed twice with PBS+/+, and then fixed with 4% PFA for 20 min at RT. Afterward, cells were incubated in a blocking buffer solution for 10 min. Samples were incubated with streptavidin-Alexa Fluor647 for 1 h at RT in the dark to selectively mark extracellular bacteria. Finally, after three washes with PBS+/+, the coverslips were transferred to glass slides, embedded in mounting medium, and sealed with nail polish. Images were acquired with a Leica AF6000LX fluorescence microscope.

## CRISPR-Cas9 mediated SynjJ1 knockout and SYNJ1 re-expression

To generate Synj1 knockout cells, $3 \times 10^4$ NIH3T3 Flp-In cells were seeded in 0.1% gelatine-coated 24-well plate. Next day, cells were transiently co-transfected with plasmids containing gRNAs targeting exon1 and exon 2 of the murine Synj1 gene (pBS-U6 sgRNA Synj1-exon1 and pBS-U6 sgRNA Synj1-exon2) together with plasmid pX459 encoding the Cas9 enzyme and containing a puromycin resistance cassette (pX459 provided by Feng Zhang; Addgene plasmid #48139). Two days after transfection, cells harboring the Cas9-encoding vector were selected using 1.5 µg/mL puromycin for 3 days, and single cells were seeded in 96-well plates and expanded in the absence of puromycin. Complete knockout of Synj1 in selected clonal lines was verified by immunoblotting against synaptojanin1 protein.

The NIH3T3 Flp-In Synj1-KO cells were co-transfected with plasmids pEF5-FRT-eGFP, pEF5-FRT-GFP-SYNJ1-WT, pEF5-FRT-GFP-SYNJ1-ΔSac1, or pEF5-FRT-GFP-SYNJ1-D730A together with pOG44 (Invitrogen; encoding the FLP recombinase) using Lipofectamine 2000 reagent. FLP recombinase mediates the stable integration of the gene of interest (GOI) by recombining the FRT sites of the plasmid with the FRT site embedded in the host cell genome. After transfection, cells were treated with hygromycin for seven days to select cells with stable integration of the FRT-cassette into the genomic locus. Next, $10^5$ cells with highest GFP fluorescence were sorted by fluorescence-activated cell sorting (FACS) and cultured in DMEM with 10% FCS, penicillin/streptomycin, and ciprofloxacin. For analysis of expression, whole cell lysates were prepared and analyzed by Western Blotting with monoclonal antibodies directed against GFP.

## Cell lysis and Western blotting

Cell lysis and western blotting were performed as described (57) with some modifications. Briefly, protein concentration was assessed using Pierce bicinchoninic assay kit (Thermo Fisher Scientific, Waltham, MA). Equal amounts of proteins were loaded on SDS-PAGE gels. Monoclonal antibody against GFP (clone JL-8) was purchased from Clontech and against Talin(8d4) antibody was from Thermo Fischer. Antibodies against synaptojanin1 (SAB2102351), Vinculin (hVIN-1) were from Sigma-Aldrich. Antibody against Paxilin (clone 165) was purchased from BD Transduction Laboratories and antibody against FAK pY397 was purchased from Biosource. Antibody against β-tubulin (clone E-7, DSHB, University of Iowa) was purified from hybridoma cell supernatants. Polyclonal rabbit antibodies against FAK (A-17), Caveolin (N-20), and c-Src (SRC2) were

from Santa Cruz Biotechnology. Goat-anti-mouse and goat-anti-rabbit IgG coupled to HRP were purchased from Jackson ImmunoResearch.

## Quantification of surface integrin expression by flow cytometry

Integrin α5 (clone 5H10-27(MFR5)) and integrin αV (clone RMV-7) antibodies were purchased from BD Biosciences. Integrin β1 (clone HMβ1-1) was obtained from Biolegend and integrin β3 antibody (clone HMb3-1) from Millipore (MA, US). The secondary antibodies [biotin-SP-conjugated goat α-rat IgG, Rhodamine Red-X-Affini-Pure goat α-Armenian hamster IgG(H + L)] and streptavidin-FITC were purchased from Jackson ImmunoResearch (West Grove, PA). For quantification of surface integrin expression, detached fibroblasts were incubated in suspension medium (DMEM containing 0.25% BSA) for 40 min at 37°C. Then, $2 \times 10^5$ cells were incubated with appropriate primary antibodies (diluted 1:300) in FACS buffer (5% heat-inactivated FCS, 1% sodium azide in PBS) for 1 hr at 4°C. After washing, secondary antibodies were applied for 1 h at 4°C, then washed again, before samples were analyzed by flow cytometry (LSRII, BD Biosciences).

## Microscopic analysis of protein recruitment to bacteria

Cells were transfected with plasmids encoding the indicated GFP-fusion proteins. One day after transfection, $1 \times 10^5$ 293 cells or $4 \times 10^4$ NIH 3T3 cells were seeded onto poly-L-lysine or gelatine-coated glass cover slips in 24-well plates. The next day, cells were infected at MOI 20 for 2 h at 37°C and 5% $CO_2$ with TAMRA-labeled *S. aureus* or *S. carnosus*. After washing with PBS+/+, samples were fixed with 4% PFA for 20 min at RT. After three washes with PBS+/+, the coverslips were mounted on glass slides. Images were acquired with a Leica AF6000LX fluorescence microscope equipped with a Leica DFC350FX camera using a 63×/1.30 HCX PL APO objective and external filter wheels for fluorescein/GFP (Exc. 470/40; Em. 535/50) or rhodamine/TAMRA (Exc. 545/30; Em. 610/75). Images were digitally processed with Image J (Wayne Rasband, National Institutes of Health, USA) and merged to yield pseudo-colored RGB pictures. For quantification, the maximum GFP fluorescence intensity at bacterial attachment sites was determined along a line scan through the attached bacteria and divided by the average GFP fluorescence intensity at a neighboring region of the same cell.

## Scanning electron microscopy

NIH 3T3 Synj1-KO cells with stable expression of GFP alone, SYNJ1-ΔSac1, SYNJ1-D730A, or SYNJ1-WT were grown on poly-lysine-coated coverslips. Confluent layers were infected with *S. aureus* at an MOI of 30 for 2 h. Then, cells were washed and fixed *in situ* with 2% glutaraldehyde/3% formaldehyde in 0.1 M cacodylate, 0.09 M sucrose, 0.01 M $CaCl_2$, and 0.01 M $MgCl_2$, pH 6.9, overnight at 4°C. Afterward, cell samples were washed with o.1 M HEPES, pH 7.0, and dehydrated with ethanol-dilutions of 1 mL [30%, 50%, 70% (overnight) 80%, 90%, and 100%] for 10 min each. After critical point drying from liquid $CO_2$, samples were sputter coated with 6 nm Pt and examined at EHT15 kV of accelerating voltage in a tungsten-wired scanning electron microscope (Zeiss Evo).

## Statistics

Infection and flow cytometry experiments were performed at least three times, and data were presented as mean ± SEM. Differences in adherence and internalization of *S. aureus* were analyzed by unpaired Student's *t* test. In all analyses, a *P* value of < 0.05 was considered to be statistically significant.

## ACKNOWLEDGMENTS

We are indebted to S. Feindler-Boeckh for excellent technical support and thankful to Tamás Balla (NIH, Bethesda, MD) for providing the GFP-tagged PLCδ-PH domain,

Reinhard Fässler (MPI for Biochemistry, Martinsried, Germany) for providing GFP-talin plasmid, and Pietro De Camilli (Yale University School of Medicine, New Haven) for providing cDNA of GFP-tagged human SYNJ1-170 and its D730A substitution mutant. We thank M. Laumann (EM Service, University of Konstanz) for help with scanning electron microscopy.

Y.S. is recipient of a fellowship from the Chinese Scholarship Council (CSC) and an associate fellow of the Graduate School Biological Sciences at the University of Konstanz. This work was supported by DFG Priority Program SPP1150 grant Ha 2856/5-1 to C.R.H.

## AUTHOR AFFILIATIONS

[1]Lehrstuhl für Zellbiologie, Universität Konstanz, Konstanz, Germany
[2]School of Life Science and Technology, Wuhan Polytechnic University, Wuhan, China
[3]Konstanz Research School Chemical Biology, Universität Konstanz, Konstanz, Germany

## AUTHOR ORCIDs

Christof R. Hauck ⬤ http://orcid.org/0000-0002-1005-2141

## FUNDING

| Funder | Grant(s) | Author(s) |
| --- | --- | --- |
| Deutsche Forschungsgemeinschaft (DFG) | Ha 2856/5-1 | Christof R. Hauck |

## AUTHOR CONTRIBUTIONS

Yong Shi, Formal analysis, Investigation, Methodology, Writing – original draft | Petra Muenzner, Formal analysis, Investigation | Stefanie Schanz-Jurinka, Investigation | Christof R. Hauck, Conceptualization, Project administration, Supervision, Writing – original draft, Writing – review and editing

## ADDITIONAL FILES

The following material is available online.

### Supplemental Material

**Supplemental figures (Spectrum02006-23-s0001.pdf).** Figures S1 and S2.

### Open Peer Review

**PEER REVIEW HISTORY (review-history.pdf).** An accounting of the reviewer comments and feedback.

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
