## [Reviewer comments · Microbiology Spectrum]

Microbiology Spectrum

The phosphatidylinositol-5' phosphatase synaptojanin1 limits integrin-mediated invasion of *Staphylococcus aureus*

Yong Shi, Petra Muenzner, Stefanie Schanz-Jurinka, and Christof Hauck

Corresponding Author(s): Christof Hauck, Universitat Konstanz Fachbereich Biologie

Review Timeline:

Submission Date:	May 12, 2023
Editorial Decision:	June 5, 2023
Revision Received:	December 14, 2023
Accepted:	January 18, 2024

Editor: Christine Josenhans

Reviewer(s): The reviewers have opted to remain anonymous.

Transaction Report:

DOI: <https://doi.org/10.1128/spectrum.02006-23>

June 5, 2023

Prof. Christof R Hauck
Universität Konstanz Fachbereich Biologie
Lehrstuhl Zellbiologie
Universitätsstrasse 10
Maildrop X908
Konstanz 78457
Germany

Re: Spectrum02006-23 (The phosphatidylinositol-5' phosphatase synaptojanin1 limits integrin-mediated invasion of *Staphylococcus aureus*)

Dear Prof. Hauck, dear Christoph:

Thank you for submitting your manuscript to Microbiology Spectrum. As you can appreciate, two experts read the manuscript in depth. They were overall positive and raised essential comments that should be addressed in your revision. When submitting the revised version of your paper, please provide (1) point-by-point responses to the issues raised by the reviewers as file type "Response to Reviewers," not in your cover letter, and (2) a PDF file that indicates the changes from the original submission (by highlighting or underlining the changes) as file type "Marked Up Manuscript - For Review Only". Please use this link to submit your revised manuscript - we strongly recommend that you submit your paper within the next 60 days or reach out to me, if you need a prolongation for the revision. Detailed instructions on submitting your revised paper are below.

Link Not Available

Sincerely,

Christine [Josenhans]

Journals Department
Reviewer comments:

Reviewer #1 (Comments for the Author):

Shi et al investigate the mechanism of *S. aureus* internalization in non-professional cells. The authors addressed specifically the role of PI-4,5-P2. They found that SYNJ1 has a negative impact on *S. aureus* internalization by host cells. Moreover, the integrin binding protein talin resulted enriched in SYNJ1 knock out cells. The introduction gives a sufficient background and justifies the aim to determine the relationship between SYNJ1 and the internalization of *S. aureus* by 293 cells and fibroblasts. The manuscript is very interesting, well written and easy to follow. However, I have some suggestions to improve the quality and

understanding of the study

1. The authors investigated the internalization of *S. aureus* by host cells by 2 different approaches: gentamicin assay and flow cytometer. However the flow cytometer-based assay of bacterial internalization is missing in the materials and methods
2. Fig. 1: Did the authors performed one statistical analysis in the fig 1 A and C?
3. The internalization of *S. aureus* within different cells is described in different figures. However, the "y" axis is different. In fig1 is CFU/cell, however, in the figures 3B and C, the results are expressed as "CFUx10E3 cells". Are these results not per cell? Moreover, the amount of intracellular CFU for cells such as SYNJ1-KO is different between the fig 3B and 5C.
4. Line 255: Do you mean Fig. 5D?
5. Discussion: do you think that this mechanism is common among all non-professional cells? Is this mechanism different in professional cells? Please include these answers in your discussion
6. Please check the sentence from line 469 "In parallel....". samples of?
7. Please check the references, some of them are in different format (ex #2)

Reviewer #2 (Comments for the Author):

Staphylococcus aureus is internalized by non-professional phagocytic cells (NPPC). This process is mediated by host integrins that bind to a fibronectin bridge, which in turn is bound by Fn-binding proteins on the staphylococcal cell wall.

In the presented study "The phosphatidylinositol-5' phosphatase synaptojanin1 limits integrin-mediated invasion of *Staphylococcus aureus*" by Shi, Muenzner, & Hauck the authors investigate the contribution of synaptojanin1 (SYNJ1) to this process.

SYNJ1 thereby was identified by an shRNA screen that encompassed ten candidate phosphatases. Upon RNAi-based knockdown of SYNJ1 *S. aureus* was internalized more efficiently by HEK293.

This was corroborated also in NIH 3T3 murine fibroblasts that were knocked-out in *Synj1* by CRISPR/Cas9-based gene editing. Complementation did recover the wild-type phenotypes and led to the reduction of *S. aureus* uptake.

A non-functional mutant as well as a deletion mutant in the Sac1 domain, however, did not complement the mutant illustrating that the activity and a Sac1 domain is needed for the observed phenotype.

The paper is generally well-written, informative and covers an interesting subject. Methods are laid out well and are experiments include the proper controls.

There are only two points that should be addressed:

- 1) Fig. 5: The subcellular distribution of the delta-Sac1 version is missing (fluorescence microscopy). It would be interesting to see if it is altered with respect to *S. aureus* localization when compared to the wild-type.
- 2) The authors focus on possible differences in the changes of talin in presence of absence of SYNJ1 activity. However, in Fig. 6 a quantification of talin recruitment is lacking. In order to corroborate their point, quantification is indispensable.

Minor Issues:

In the Methods Section:

- a) Addgene sources of pLKO1, pMD2.G and pPAX2 are not listed.
- b) It should be noted somewhere, that the SYNJ1 shRNA was directed against the 3' UTR in the last exon, which allows for the observed complementation in the KD cell lines.
- c) The section on fluorescence microscopy (starting at line 474) should list the channels (Ex/Em) and further details of image acquisition.

General:

- d) line 469: 'which resemble the "recovered intracellular bacteria"' : Do you mean: "which constitute" or similar?
- e) Whenever the mouse homolog of human SYNJ1 is used, the correct gene symbol (capitalization) *Synj1* should be used.
- f) line 213: "exaggerated endocytosis" : probably more suitably is "enhanced endocytosis" or similar.

Staff Comments:

Preparing Revision Guidelines

Please return the manuscript within 60 days; if you cannot complete the modification within this time period, please contact me. If you do not wish to modify the manuscript and prefer to submit it to another journal, please notify me of your decision immediately so that the manuscript may be formally withdrawn from consideration by Microbiology Spectrum.

Editor and Reviewer comments - Response_to_Reviewers:

Reviewer #1 (Comments for the Author):

Shi et al investigate the mechanism of *S. aureus* internalization in non-professional cells. The authors addressed specifically the role of PI-4,5-P2. They found that SYNJ1 has a negative impact on *S. aureus* internalization by host cells. Moreover, the integrin binding protein talin resulted enriched in SYNJ1 knock out cells. The introduction gives a sufficient background and justifies the aim to determine the relationship between SYNJ1 and the internalization of *S. aureus* by 293 cells and fibroblasts. The manuscript is very interesting, well written and easy to follow. However, I have some suggestions to improve the quality and understanding of the study.

1. The authors investigated the internalization of *S. aureus* by host cells by 2 different approaches: gentamicin assay and flow cytometer. However the flow cytometer-based assay of bacterial internalization is missing in the materials and methods

We apologize for this omission and we have now included not only a description of the flow-cytometer-based quantification of bacterial internalization, but also a detailed description of the fluorescence labeling procedure of the bacteria. Furthermore, we refer now to our prior publications, where we present these methods and describe the controls used to establish these procedures in more detail.

The experimental approach and the reagents for bacterial labeling can be found in the Material & Methods section on page 14, line 400ff:

“In certain assays, PBS-washed staphylococci from log-phase cultures were labeled with 1 µg/ml 5-(6)-carboxy-fluorescein-succinimidylester (CFSE), 5-(and-6)-carboxy-tetramethyl-rhodamine-succinimidylester (TAMRA-SE), Pacific Blue-succinimidylester (PBSE), or Sulfo-NHS-LC-biotin (all purchased from ThermoFisher, Life Technologies, Darmstadt, Germany) for 20 min at 37°C prior to infection as previously described (14,52,53).”

The description of flow cytometry-based evaluation of bacterial internalization can now be found in the Material& Methods section from page 16, line 477ff:

“Evaluation of bacterial internalization by flow cytometry

For flow cytometric analysis of bacterial uptake, cells were infected with CFSE-labelled bacteria and analyzed as described (53). Briefly, 4×10^5 HEK293 cells or 1×10^5 fibroblasts were seeded one day before infection in poly-L-lysine or gelatine-coated 6-well plates. PBS-washed bacteria from logarithmic growing cultures were stained with 5-(6)-carboxyfluorescein-succinimidylester (CFSE) and added at a multiplicity of infection (MOI) of 20. After 2 hours, infected cells were detached, washed with PBS and cell-associated

fluorescein fluorescence was analysed by flow cytometry. To quench signals from extracellular bacteria, trypan blue solution (0.4%; Sigma, Taufkirchen, Germany) was added to a final concentration of 0.2% directly before analysis. Samples were analysed on a LSRII (BD Biosciences) by gating on the eukaryotic cells based on forward and side scatter and cell-associated fluorescence of 10,000 cells per sample was measured in fluorescence channel 1 (FL1-H) detecting CFSE fluorescence as a measure of internalized bacteria.”

2. Fig. 1: Did the authors performed one statistical analysis in the fig 1 A and C?

Though the experimental approach depicted in Figure 1A and C was repeated multiple times ($n = 4 - 6$), we did not perform a statistical evaluation. On the one hand, this was due to the variable outcome observed with regard to cell-associated and intracellular bacteria, which we clearly depict in Fig. 1A and in Suppl. Fig. S1A and S1B. The variable outcome might reflect differences in knock-down efficiencies between biological replicates, but also the potential additional effect of phosphatase knock-down on receptor surface expression as well as on intracellular trafficking (and therefore survival) of internalized bacteria, which we have already mentioned in the results section.

On the other hand, this first approach was intended to serve as an initial screen (and not a definitive proof) to identify potential candidate enzymes for a more detailed, consecutive analysis. Therefore, we have focussed the remainder of the study on verifying the gain-of-function phenotype observed for Synптоjanin-1-knock-down cells with independent genetic and functional analyses, rather than spending additional time and resources on determining statistical significance of an initial screen.

3. The internalization of *S. aureus* within different cells is described in different figures. However, the "y" axis is different. In fig1 is CFU/cell, however, in the figures 3B and C, the results are expressed as "CFUx10E3 cells". Are these results not per cell? Moreover, the amount of intracellular CFU for cells such as SYNJ1-KO is different between the fig 3B and 5C.

The reviewer is completely correct in that the different experimental approaches used to evaluate intracellular bacteria lead to different labeling of Y-axes in these Figures. E.g. in Fig. 2D and 2E, the number of intracellular bacteria is given as bacteria/cell, as in these cases the intracellular bacteria were quantified on the basis of immunofluorescence stainings of intracellular bacteria, which are counted on a cell-based basis. In other cases, the intracellular bacteria were determined by antibiotic protection assays (gentamicin protection assays), where the infected cell culture is lysed and the evaluation is based on the number of viable bacteria (cfu) recovered from the whole sample (e.g. Fig. 3B). This explains the different labeling of the y-axes.

However, we are thankful that the reviewer alerted us to the gross differences in cfu recovered from SYNJ1-KO cells in Fig. 3B and Fig. 5C. In this case, we have inadvertently used the wrong scale as the numbers on the y-axis reflected $\text{cfu} \times 10^3$ instead of $\text{cfu} \times 10^4$ (left panel) and $\text{cfu} \times 10^2$ instead of $\text{cfu} \times 10^3$ (right panel) in Fig. 5C. We have now corrected this mistake and now the numbers of recovered intracellular bacteria from SYNJ1 cells are

similar in these two experiments, while the number of total extracellular bacteria differs by a factor of ~4 between Fig. 3B and Fig. 5C. We do not have a definite answer for this difference, but we would like to stress that the fibroblasts used in Fig. 3B are the Control knock-out and the SYNJ1 knock-out cells, while the cells used in Fig. 5B are the complemented cells (expressing GFP or GFP-fused SYNJ1 proteins), which are all derived from the SYNJ1 ko cells. Therefore, the cell lines used in these two experiments are related, but not identical. In addition to slight differences in the density of the bacterial inoculum between experiments, the genetic differences between the knock-out cells and the GFP-complemented knock-out cells might contribute to the increased amount of total cell-associated bacteria found in Fig. 5C.

4. Line 255: Do you mean Fig. 5D?

At this point of the results section, we indeed referred to Fig. 5C as written in the text, where results with cells re-expressing GFP-synaptojanin-1- Δ Sac1 are presented.

5. Discussion: do you think that this mechanism is common among all non-professional cells? Is this mechanism different in professional cells? Please include these answers in your discussion

We thank the reviewer for this advice. In the discussion section on page 11, line 314ff we refer to the ubiquitous expression of synaptojanin1 and emphasize our observation of synaptojanin1-dependent effects in different cell types (murine fibroblasts, human 293 kidney epithelial cells) suggesting that this process might play a role in multiple cell types.

“Here, we report that the ubiquitously expressed PI-4,5-P₂-directed lipid phosphatase synaptojanin-1 counteracts the internalisation of integrin-binding Staphylococcus aureus. Indeed, deficiency of synaptojanin-1 leads to a gain-of-function with regard to S. aureus invasion into different non-phagocytic cell types accompanied by increased local levels of PI-4,5-P₂ and elevated recruitment of the PI-4,5-P₂-binding integrin activator talin. These findings suggest that the lipid phosphatase Synaptojanin-1 not only limits the integrin-mediated uptake of pathogenic bacteria, but might also regulate integrin-dependent processes in different cell types under physiological conditions.”

A role for FnBP-fibronectin-integrin-beta1-mediated uptake of *S. aureus* by professional phagocytic cells has been reported before (Shinji H et al. (1998) Different effects of fibronectin on the phagocytosis of Staphylococcus aureus and coagulase-negative staphylococci by murine peritoneal macrophages. Microbiol Immunol. 42(12): 851-61. doi: 10.1111/j.1348-0421.1998.tb02361.x.; Shinji H et al. (2003) Fibronectin bound to the surface of Staphylococcus aureus induces association of very late antigen 5 and intracellular signaling factors with macrophage cytoskeleton. Infect Immun. 71(1):140-6. doi: 10.1128/IAI.71.1.140-146.2003; Wang QQ et al. (2008) Integrin beta 1 regulates phagosome maturation in macrophages through Rac expression. J Immunol. 180(4):2419-28. doi: 10.4049/jimmunol.180.4.2419.). However, additional receptors on professional phagocytes such as complement-, scavenger- or Fc γ -receptors also contribute to the internalization of *S.*

aureus depending on the genetic make-up of the pathogens and/or the presence of host opsonins (Negrón O et al. (2022) Fibrin(ogen) engagement of *S. aureus* promotes the host antimicrobial response and suppression of microbe dissemination following peritoneal infection. PLoS Pathog. 18(1):e1010227. doi: 10.1371/journal.ppat.1010227; Lukácsi S et al. (2017) The role of CR3 (CD11b/CD18) and CR4 (CD11c/CD18) in complement-mediated phagocytosis and podosome formation by human phagocytes. Immunol Lett. 189:64-72. doi: 10.1016/j.imlet.2017.05.014). Therefore, it will require meticulous experimentation to sort out, if synaptojanin1 has a major role in all or only in a selected subset of these endocytotic processes in professional phagocytes such as macrophages or granulocytes. The data presented in our manuscript only cover non-professional phagocytes and we feel that we would stretch our results too far if would extrapolate to phagocytosis mediated by macrophages or granulocytes. Therefore, we have refrained from including this aspect in the discussion section.

6. Please check the sentence from line 469 "In parallel...". samples of?

This description of the gentamicin protection assay should make clear, that similarly infected cells were processed in two distinct ways: i) incubated with gentamicin to kill extracellular bacteria before lysis of the infected host cells and release of viable intracellular bacteria and ii) lysis of infected host cells without prior gentamicin treatment to release cell-associated (attached) and intracellular bacteria (= total cell-associated bacteria). These two samples are infected in parallel, using the same set of transfected cells and the same bacterial inoculum. We have now re-written this section to clarify this point. This paragraph on page 16, line 477ff now reads:

“Gentamicin protection assay

2x10⁵ 293 cells or 5x10⁴ NIH 3T3 cells were seeded into poly-L-lysine coated 24-well plates. Cells were infected at MOI 20 for 2 h at 37°C and 5% CO₂. Two sets of triplicate wells with identically infected cells were processed in two different ways: A) to evaluate the number of viable intracellular bacteria, the medium was carefully replaced with a fresh medium containing 50 µg/ml gentamicin. After incubation for 1 hr at 37°C intracellular bacteria were released by incubation with 0.5% saponin for 15 min at 37°C. Released bacteria were diluted in PBS and plated on TSB agar plates to determine the colony-forming units (cfu). Colony counts of these samples are referred to as “recovered intracellular bacteria”. B) to determine the total number of cell-associated intracellular and extracellular bacteria, the infected cells were gently washed with PBS, lysed with 0.5% saponin without prior gentamicin treatment and dilutions were plated on TSB agar plates. Colony counts of these samples are referred to as “total cell-associated bacteria”.

7. Please check the references, some of them are in different format (ex #2)

The reviewer is correct in pointing out differences in reference format and we have corrected this mistake.

Reviewer #2 (Comments for the Author):

Staphylococcus aureus is internalized by non-professional phagocytic cells (NPPC). This process is mediated by host integrins that bind to a fibronectin bridge, which in turn is bound by Fn-binding proteins on the staphylococcal cell wall.

In the presented study "The phosphatidylinositol-5' phosphatase synaptojanin1 limits integrin-mediated invasion of *Staphylococcus aureus*" by Shi, Muenzner, & Hauck the authors investigate the contribution of synaptojanin1 (SYNJ1) to this process.

SYNJ1 thereby was identified by an shRNA screen that encompassed ten candidate phosphatases. Upon RNAi-based knockdown of SYNJ1 *S. aureus* was internalized more efficiently by HEK293.

This was corroborated also in NIH 3T3 murine fibroblasts that were knocked-out in *Synj1* by CRISPR/Cas9-based gene editing. Complementation did recover the wild-type phenotypes and led to the reduction of *S. aureus* uptake.

A non-functional mutant as well as a deletion mutant in the Sac1 domain, however, did not complement the mutant illustrating that the activity and a Sac1 domain is needed for the observed phenotype.

The paper is generally well-written, informative and covers an interesting subject. Methods are laid out well and the experiments include the proper controls.

There are only two points that should be addressed:

1) Fig. 5: The subcellular distribution of the delta-Sac1 version is missing (fluorescence microscopy). It would be interesting to see if it is altered with respect to *S. aureus* localization when compared to the wild-type.

We did not have this GFP-SYNJ1- Δ Sac1 construct at the beginning of our study, when we analysed the subcellular localization of the wildtype and the phosphatase dead mutant in *S. aureus*-infected 293T cells (Fig. 2F). We have now repeated these experiments with transiently transfected 293T cells expressing GFP-SYNJ1- Δ Sac1. As expected, the GFP-SYNJ1- Δ Sac1 mutant shows a similar distribution as the phosphatase inactive GFP-SYNJ1-D730A mutant, with only slightly elevated levels surrounding the bacteria. This is in contrast to the wildtype GFP-SYNJ1, which is strongly enriched in a vesicular pattern around cell-associated bacteria. For the mutant without Sac1 domain we find a more broad, cytosolic distribution indicating that the Sac1 domain aids in recruiting SYNJ1 to the inner leaflet of cellular membranes. This result is now included in Fig. 2F and this observation is reported in the "Results" section on page 6, line 169ff:

"To investigate if SYNJ1 acts in the vicinity of bacteria during the internalization process, we expressed GFP and GFP-SYNJ1-WT and infected the cells with the integrin-binding S. aureus or the non-pathogenic S. carnosus, which does not engage integrins (Fig. 2F). SYNJ1-WT, but not GFP alone, was enriched around S. aureus during the internalization

process and contact between non-integrin-binding S. carnosus and the cell membrane, which only occurred in rare cases, did not result in a redistribution of SYNJ1 (Fig. 2F). In addition to a prominent staining on the membrane surrounding S. aureus, wildtype SYNJ1 was also concentrated on vesicles in the vicinity of the bacteria (Fig. 2F). We also tested the localization of the phosphatase-inactive GFP-SYNJ1-D730A and of synaptojanin1 lacking the Sac1 domain (GFP-SYNJ1-ΔSac1). In both cases, these proteins showed a more diffuse distribution in the cytosol of the cell and did not strongly enrich around cell-associated bacteria (Fig. 2F). Together, these findings provide evidence that SYNJ1 co-localizes with S. aureus during entry and further demonstrate that the activity of this enzyme limits the integrin-mediated internalization of S. aureus into non-professional phagocytes.”

2) The authors focus on possible differences in the changes of talin in presence of absence of SYNJ1 activity. However, in Fig. 6 a quantification of talin recruitment is lacking. In order to corroborate their point, quantification is indispensable.

Based on this important suggestion by the reviewer, we have quantified the recruitment of GFP-talin at individual bacterial attachment sites. We relate the maximum GFP-talin intensity at these S. aureus-attachment sites to neighbouring sites of the identical cell, where no bacteria are in contact with the cell. In that way, the differences in GFP-talin expression between individual transfected cells, which are unavoidable in such a transient transfection setup, do not confound the analysis. By comparing the GFP-talin recruitment in Synj1-KO cells versus the GFP-talin recruitment in Ctrl KO cells, we find that GFP-talin recruitment to S. aureus attachment sites is significantly increased upon deletion of the lipid phosphatase supporting our initial conclusions. These additional data are now added as novel Fig. 6C and are accompanied by an additional description in the results section on page 10, line 286ff:

“However, in Synj1-KO cells a strong enrichment of GFP-talin around the cell-associated pathogens was observed (Fig. 6A). The distribution of talin was not influenced by the occasional contact of non-pathogenic S. carnosus with Synj1-KO cells (Fig. 6B). Quantification of GFP-intensity through bacterial attachment sites indicated a ~30% increase of GFP-talin recruitment in Synj1-KO cells compared to Ctrl KO cells (Fig. 6C).”

Minor Issues:

In the Methods Section:

a) Addgene sources of pLKO1, pMD2.G and pPAX2 are not listed.

Please excuse this omission. We have added this information in the Material & Methods section to now read on page 16, line 467ff:

“The lentiviral vectors pLKO.1-TRC (#10878), pMD2.G (##12259) and psPAX2 (#12260) were purchased from Addgene.”

b) It should be noted somewhere, that the SYNJ1 shRNA was directed against the 3' UTR in the last exon, which allows for the observed complementation in the KD cell lines.

We thank the reviewer for this valuable advice. We have added this information now on page 6, line 154ff with the following sentence:

“The complementation of the SYNJ1 knock-down cells with constructs harboring the synaptotagmin-1 cDNA was possible, since the SYNJ1 shRNA-oligonucleotide targeted the 3' untranslated region of the human synaptotagmin mRNA. GFP and the GFP-fusion proteins were expressed at equivalent levels in the transfected SYNJ1 knockdown cells as confirmed by Western blotting (Suppl. Fig. S1C).”

c) The section on fluorescence microscopy (starting at line 474) should list the channels (Ex/Em) and further details of image acquisition.

We have expanded this paragraph in the Material&Methods section to include the requested information. This paragraph on page 19, line 579ff now reads:

“Images were acquired with a Leica AF6000LX fluorescence microscope equipped with a Leica DFC350FX camera using a 63x / 1.30 HCX PL APO objective and external filter wheels for fluorescein/GFP (Exc. 470/40; Em. 535/50) or rhodamine/TAMRA (Exc. 545/30; Em. 610/75). Images were digitally processed with Image J (Wayne Rasband, National Institutes of Health, USA) and merged to yield pseudo-colored RGB pictures.”

d) line 469: 'which resemble the "recovered intracellular bacteria"' : Do you mean: "which constitute" or similar?

Also prompted by the comments of Reviewer1, we have now re-written this section to clarify this point. This paragraph on page 16, line 477ff now reads:

“Gentamicin protection assay

2x10⁵ 293 cells or 5x10⁴ NIH 3T3 cells were seeded into poly-L-lysine coated 24-well plates. Cells were infected at MOI 20 for 2 h at 37°C and 5% CO₂. Two sets of triplicate wells with identically infected cells were processed in two different ways: A) to evaluate the number of viable intracellular bacteria, the medium was carefully replaced with a fresh medium containing 50 µg/ml gentamicin. After incubation for 1 hr at 37°C intracellular bacteria were released by incubation with 0.5% saponin for 15 min at 37°C. Released bacteria were diluted in PBS and plated on TSB agar plates to determine the colony-forming units (cfu). Colony

counts of these samples are referred to as "recovered intracellular bacteria". B) to determine the total number of cell-associated intracellular and extracellular bacteria, the infected cells were gently washed with PBS, lysed with 0.5% saponin without prior gentamicin treatment and dilutions were plated on TSB agar plates. Colony counts of these samples are referred to as "total cell-associated bacteria".

e) Whenever the mouse homolog of human SYNJ1 is used, the correct gene symbol (capitalization) Synj1 should be used.

We thank the reviewer for this valuable advice and have modified the text throughout the manuscript accordingly.

f) line 213: "exaggerated endocytosis" : probably more suitably is "enhanced endocytosis" or similar.

We have altered the phrase according to the reviewer's suggestion to now read on page 8, line 218ff:

"Therefore, we speculated that the lack of Synj1 might promote elevated levels of PI-4,5-P₂ at bacterial attachment sites leading to increased endocytosis of the bacteria."

Re: Spectrum02006-23R1 (The phosphatidylinositol-5' phosphatase synaptojanin1 limits integrin-mediated invasion of *Staphylococcus aureus*)

Dear Prof. Christof R Hauck, dear Christof:

Your revised manuscript has been accepted, and I am forwarding it to the ASM production staff for publication. Your paper will first be checked to make sure all elements meet the technical requirements. ASM staff will contact you if anything needs to be revised before copyediting and production can begin. Otherwise, you will be notified when your proofs are ready to be viewed.

Sincerely,
Christine Josenhans
Editor
Microbiology Spectrum